# FedSynth: Gradient Compression via Synthetic Data in Federated Learning

## Abstract

Model compression is important in federated learning (FL) with large models to reduce communication cost. Prior works have been focusing on sparsification based compression that could desparately affect the global model accuracy. In this work, we propose a new scheme for upstream communication where instead of transmitting the model update, each client learns and transmits a light-weight synthetic dataset such that using it as the training data, the model performs similarly well on the real training data. The server will recover the local model update via the synthetic data and apply standard aggregation. We then provide a new algorithm `FedSynth` to learn the synthetic data locally. Empirically, we find our method is comparable/better than random masking baselines in all three common federated learning benchmark datasets.

## 1 Introduction

Federated Learning(FL) has been widely studied recently to train machine learning models without directly accessing users' data. Despite being successful in achieving high utility performance compared to centralized training, huge communication costs induced by current FL algorithms like FedAvg[10] prevents using federated data to train large scale models. Specifically, communicating the entire model between each client and the server could drastically slower the training process, and imposes communication costs on the users. Many prior efforts have focused on sparsifying the model to reduce communication cost, including masking the model updates[7, 14], low precision training with quantization[1, 2, 3, 4, 12, 15], distillation[11, 18], etc. However, sparsification based methods usually suffers from communication cost-utility trade-off: high compression rate could hurt model quality.

To overcome such limitation, we propose a different way to think about model compression in FL. Instead of a model update, which is the same size as the original model, we propose sending a batch of carefully optimized synthetic data, which is significantly smaller in size. Each client crafts a set of synthetic data such that the model updated by the synthetic data performs well on the client's original training data. In this way, we could use a dataset that is significantly smaller than the client training set to obtain a similar model update as if we use the original data. Each client then send the synthetic data to the server. Upon receiving the synthetic data from each client, the server will use it to recover the model updated by synthetic data.

Having this intuition in mind, we formally propose a new objective in federated learning for local clients at each communication round. Using this formulation, we propose an effective solver that could adapt to a wide family of existing federated learning algorithm. Specifically, we develop an algorithm that learns synthetic data for local clients under the FedAvg[10] framework. We empirically evaluate and compare our method with prior works, demonstrating advantage of transmitting synthetic data as an effective compression technique.

Submitted to 36th Conference on Neural Information Processing Systems (NeurIPS 2022). Do not distribute.

## 2  Background and Related Work

**Compression in FL**   Model compression techniques has been widely studied in machine learning community with a centralized dataset. Some widely used methods include gradient quantization[1], gradient ternarization[17], using the sign of gradient[3], pruning based methods like masking[4], etc. In federated learning, compression could happen at two places: transmitting model updates from local client to the server (upstream); transmitting updated global model from the server to local client (downstream). Recent works have proposed using sparsification techniques for both upstream and downstream communication to reduce the cost of large scale federated learning [9, 12, 13]. Different from our work, these works focused on communicating a sparse model between server and client, where the model performance could significantly degrade given the same number of training steps. Goetz and Tewari [8] proposed a similar scheme to transmit synthetic data via upstream communication. However, they propose optimizing synthetic data to minimize the distance between model updated by synthetic data and model updated by training data. Our proposed objective directly optimizes the synthetic data so that the resulting model achieves good performance on the true training data.

**Dataset Distillation**   A motivation of our work is to learn a small set of synthetic data that could perform equally well on a given model compared to real data used to train the model. [16] proposed a dataset distillation algorithm that optimizes synthetic data $w_{syn}$ such that the model learned using $w_{syn}$ as the training data approximates the model learned using the true training data. Although similar to our approach, they only considered distillation from a centralized dataset at one time while in our case we learn the objective locally for each client at *every* communication round. Our proposed method is also more general in the method to update the model using synthetic data (See Section 3.2) rather than restricted to SGD.

## 3  Communication via Synthetic Data

### 3.1  Formulation

Traditional Federated Learning(FL) aims at solving the following objective:

$$\min_w \sum_{k=1}^{K} p_k F_k(w) \tag{1}$$

where $F_k(w)$ is the local objective for client $k$, $p_k$ are pre-defined weights such that $\sum_k p_k = 1$. At each communication round, the central server selects a subset of clients and send the current model to the them. Each client then separately optimizes its local objective iteratively using stochastic gradients. Then the server collects and aggregates the model updates from every client to obtain the new global model. Note that model updates have the same size as the actual global model, which means if a large-scale model is used as the model, the client would need to send a model as large as the global model. Under our proposed method, instead of sending the model updates, each client now sends batches of synthetic data generated locally to the server. The server will then utilize the synthetic data to recover the local model updates. We formalize the optimization process of synthetic data as the following.

For client $k$, let $D_k^{tr} = (X_k, Y_k)$ be the training data and $w_k^t$ the local copy of global model at the $t$-th communication round. In traditional FL, at every communication round, client $k$ tries to solve

$$\min_w F_k(D_k^{tr}; w) \tag{2}$$

using $w_k^t$ is as the initialization of $w$. Note that a lot of existing federated learning methods rely on using iterative gradient methods to solve for Equation 2. Let's define the update process for any client $k$ at communication round $t$ to be $\texttt{ClientUpdate}^k(\cdot; w_k^t)$. Thus, in the traditional FL setting, local optimization process could be written as $\texttt{ClientUpdate}^k(D_k^{tr}; w_k^t)$. The goal is to find a set of synthetic data $D_k^{syn} = \{x_k^i, y_k^i\}_{i=1,\cdots,m}$ that is significantly smaller in memory size than $w$, such that $\texttt{ClientUpdate}^k(D_k^{syn}; w_k^t)$ is similar to optimizing $\texttt{ClientUpdate}^k(D_k^{tr}; w_k^t)$. At the end of that communication round, client $k$ will send $D_k^{syn}$ to the server and the server could recover $w_k^{syn} = \texttt{ClientUpdate}^k(D_k^{syn}; w_k^t)$ and utilizes $w_k^{syn}$ as client $k$'s updated model for aggregation.

Now the problem becomes how can we find $D_k^{syn}$ that distills the knowledge from $D_k^{tr}$. The most direct way to do so is to minimize a distance metric between the model generated from synthetic data and the model generated from true train data. However, note that our purpose is that using $D_k^{syn}$, we could obtain an updated model $w_k^{syn}$ such that $F_k(D_k^{tr}; w_k^{syn})$ is as good as $F_k(D_k^{tr}; w_k^{tr})$. With certain purpose in mind, we propose the following objective:

$$\min_{D_k^{syn}} F_k \left( D_k^{tr}; \arg\min_w F_k \left( D_k^{syn}; w \right) \right) \tag{3}$$

Note that when Equation 2 is convex in $w$ and Equation 3 is convex in $D_k^{syn}$, given the same $D_k^{tr}$, both equations are essentially finding the same optimal local model. However, when there doesn't exist a closed form solution for $\arg\min_w F_k(D_k^{syn}; w)$, the inner optimization problem for Equation 3 could not be solved exactly with finite number of steps at every communication round. To find an approximate solution, most existing FL methods utilize gradient based methods like SGD. Therefore, we propose to optimize the following objective in practice instead of Equation 3:

$$\min_{D_k^{syn}} F_k \left( D_k^{tr}; \texttt{ClientUpdate}_k \left( D_k^{syn}; w_k^t \right) \right) \tag{4}$$

Without loss of generality, $\texttt{ClientUpdate}$ could be any local optimization methods including GD, SGD, etc.

## 3.2 Algorithm

We summarize our algorithm for federated learning via synthetic data in Algorithm 1. Our algorithm is based off of FedAvg[10], a communication efficient method widely used in federated learning. At each communication round, instead of performing SGD on the local training data, each selected client $k$ first initializes a synthetic dataset $D_k^{syn}$ (line 5). To find the best $w_k^{syn}$, synthetic updated model generated by $D_k^{syn}$ (line 7) that minimizes the its loss on the original training data $D_k^{tr}$, we propose to apply gradient descent on $D_k^{syn}$ for multiple iterations (line 8). After that, client $k$ would send $D_k^{syn}$, an entity that requires significantly less storage compared to the model weight, back to the server. To recover client $k$'s learned model, the server updates $w_k^t$ with $D_k^{syn}$ using the same process client $k$ generated $w_k^{syn}$ (line 12). We also provide an example of the $\texttt{ClientUpdate}$ method: running SGD on $w_k^t$ using the $D_k^{syn}$ (line 16-19). This is consistent to the local update rule in FedAvg, where client applies SGD to update $w_k^t$ using its local training data. In order to fully utilize the advantage of using synthetic data to distill the information from the original training data, we also propose the following techniques while learning $D_k^{syn}$.

**Multiple batches of synthetic data** At every communication round, FedAvg allows a selected client $k$ to split its local training data into multiple batches and perform minibatch-SGD for multiple epochs. Motivated by this, we allow client $k$ to create multiple batches of synthetic data. Instead of running one step gradient descent on the entire synthetic data, client $k$ updates $w_k^t$ sequentially using different batches of synthetic data, as specified in Line 18 of Algorithm 1.

**Trainable label** In a traditional supervised classification task, the data usually has fixed label $y$. When using cross entropy as the loss function, fixed $y$ is encoded as an one-hot vector in $\mathbb{R}^{|C|}$ where $C$ is the set of all labels. However, this is not necessary for synthetic data. The purpose of using synthetic data is only to generate a model that performs well on the real training data. Restricting any synthetic $x_i$ to have a fixed label $y_i$ is too stringent and limit the search space for pairs of $(x_i, y_i)$ to learn the information of the original training data. Hence, we propose randomly initializing $y_i \sim Uniform(0, 1)^{|C|}$. While updating synthetic data $(x_i, y_i)$, we calculate

$$x_i \leftarrow x_i - \eta_x \nabla_{x_i} F_k(D_k^{tr}; w_k^{syn}) \tag{5}$$

$$y_i \leftarrow y_i - \eta_y \nabla_{y_i} F_k(D_k^{tr}; w_k^{syn}) \tag{6}$$

It is worth noting that under certain scenario, we do not limit $y_i$ to be a vector representing the probability that $x_i$ belongs to a certain class. Each entry for $y_i$ could be arbitrary real numbers so that we could search in the entire $\mathbb{R}^{|C|}$ to find a good local minima for $y_i$.

## 4 Experiments

In this section we empirically evaluate our Algorithm 1 on common large scale federated learning benchmarks. We first demonstrate that given the same compression rate our method could achieve

---
**Algorithm 1** FedSynth
---
1: **Input:** $T$, $E$, $\eta$, $\eta_w$, $w^0$, $\{D_k^{tr}\}_{k=1,\cdots,K}$
2: **for** $t = 0, \cdots, T-1$ **do**
3:     Server selects a subset of clients $S_t$ and broadcasts $w^t$ to $S_t$.
4:     **for all** $k \in S_t$ in parallel **do**
5:         Client $k$ initializes $w_k^t = w^t$ and $m$ batches of synthetic data $D_k^{syn} = \{x_i, y_i\}_{i=1,\cdots,m}$.
6:         **for** $j = 0, 1, \cdots, E$ **do**
7:             Client $k$ obtains the model updated by $D_k^{syn}$
$$w_k^{syn} = \texttt{ClientUpdate}(D_k^{syn}; w_k^t)$$

8:             Client $k$ updates $D_k^{syn}$ by
$$D_k^{syn} \leftarrow D_k^{syn} - \eta \nabla_{D_k^{syn}} F_k(D_k^{tr}; w_k^{syn})$$
9:         **end for**
10:        Client $k$ sends $D_k^{syn}$ back to the server.
11:     **end for**
12:     Server recovers $\widehat{w}_k^{syn} = \texttt{ClientUpdate}(D_k^{syn}, w_k^t)$ for every $k$.
13:     Server aggregates the weight

$$w^{t+1} = w^t + \frac{1}{|S_t|} \sum_{k \in S_t} (\widehat{w}_k^{syn} - w^t)$$

14: **end for**
15: **return** $w^T$

---
16: $\texttt{ClientUpdate}(\{x_i, y_i\}_{i=1,2,\cdots,m}; w)$
17: **for** $j = 1, \cdots, m$ **do**
18:     Client performs minibatch-SGD locally
$$w \leftarrow w - \eta_w \nabla_w F_k((x_i, y_i); w)$$
19: **end for**
---

higher test accuracy then random masking, a popular compression method used in federated learning. We also show how number of batches of synthetic data($m$) and the size of each batch could affect the resulting model's performance.

## 4.1 Experimental setup details

For fair comparison, all methods are trained for the same amount of communication rounds for each dataset. For baseline method(Random Masking), we only apply the compression technique during the upstream communication (i.e. only compress the model sent from client to the server) in order to be consistent to our method. For a fixed compression rate, we apply grid search to tune the hyperparameters($E$, $\eta$, $\eta_w$) for our method on the validation data and report the test accuracy corresponding to the best validation accuracy. We similarly finetune the hyperparameters for random masking and baseline FedAvg as well. For all our experiments, we evaluate the test accuracy and compression rate for a fixed number of communication rounds $T$. All experiments are performed on common federated learning benchmark datasets. Data for FEMNIST and Reddit are naturally partitioned among all the users.

## 4.2 Comparison between FedSynth and Random Masking

We evaluated our method on three commonly used federated benchmark datasets: FEMNIST, MNIST, and Reddit [5]. The results are shown in Table 1. Under all three datasets, our method achieves comparable/better performance then baseline random masking methods. Specifically, under a low compression rate, there is an advantage of using our method over random masking in all three datasets. Under Reddit next word prediction task where the utility performance is extremely sensitive to masking, our method that utilizes synthetic data without trainable $y$ achieves higher test accuracy than prior works under all compression rates we experimented. It is also worth noting that our method with trainable $y$ does not always outperform synthetic data with a fixed label, as shown in the FEMNIST experiment.

**Table 1:** Comparing our method with previous compression baselines. The best performance under each given compression rate is **highlighted**.

| FEMNIST | FedAvg | Random Masking | FedSynth(Ours) | FedSynth w/ Trainable $y$ (Ours) |
|---|---|---|---|---|
| 1x | 69.29 | 69.29 | 69.29 | 69.29 |
| 5.8x | - | 68.21 | **68.63** | 46.67 |
| 11.6x | - | **67.34** | 63.27 | 39.98 |
| **MNIST** | FedAvg | Random Masking | FedSynth(Ours) | FedSynth w/ Trainable $y$ (Ours) |
| 1x | 97.74 | 97.74 | 97.74 | 97.74 |
| 7.8x | - | 97.08 | 95.28 | **97.25** |
| 15.6x | - | **96.94** | 93.68 | 96.62 |
| **Reddit** | FedAvg | Random Masking | FedSynth(Ours) | FedSynth w/ Trainable $y$ (Ours) |
| 1x | 14.19 | 14.19 | 14.19 | 14.19 |
| 1.3x | - | 8.20 | **8.86** | - |
| 2.6x | - | 4.87 | **4.89** | - |

**Table 2:** The effect of number of synthetic batches and batch size on the test accuracy of FedSynth.

| **FEMNIST** | | FedSynth | FedSynth w/ Trainable $y$ |
|---|---|---|---|
| Synthetic Batches | Batch size | | |
| 1 | 10 | 11.87 | 10.93 |
| 5 | 5 | 64.21 | 29.43 |
| 10 | 5 | 67.92 | 46.67 |
| 10 | 2 | 60.19 | 39.72 |
| 20 | 2 | 68.63 | 42.32 |
| 10 | 1 | 56.47 | 26.29 |
| 20 | 1 | 62.37 | 39.98 |

## 4.3 Number of synthetic batches vs. batch size

As we mentioned in Section 3, each client could split their synthetic data into multiple batches. In Table 2, we demonstrate how different number of synthetic batches and batch size could influence the model performance. Given the same number of data points, having more small batches significantly outperforms having few large batches. On one extreme where we treat the entire synthetic dataset as a large batch, i.e. $w_k^t$ is only updated once to get $w_k^{syn}$, model trained using our methods is barely useful. However, on the other extreme where every single piece of data is treated as a separate batch, our method is able to achieve significantly better performance. We would also like to highlight that the more synthetic data we use, the better model performance we could obtain, given the same number of synthetic batches used for getting $w_k^{syn}$.

## 5 Conclusion and Future works

In this work, we propose a new objective for communication efficient federated learning along with a practical algorithm to solve it. We showed empirically that our methods outperforms baseline methods at low compression level in all three datasets we evaluated. In future works, we aim at making the algorithm more scalable so that learning of synthetic data would require less iterations. We also want to look at sending differentially private synthetic data to protect the local data from potential privacy leakage.

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

**Table 3**

| Dataset | Number of clients | Model | Task Type |
|---|---|---|---|
| FEMNIST [5, 6] | 1000 | 4-layer CNN[19] | 62-class image classification |
| MNIST | 60 | 4-layer CNN[19] | 10-class image classification |
| Reddit [5] | 100 | Stacked LSTM | Next word precition |

# A   Appendix

## A.1   Datasets and Models

We summarize the details of the datasets and models we used in our empirical study in Table 3. Our experiments include both text (Reddit) and image (MNIST and FEMNIST) datasets with both classification task (MNIST and FEMNIST) and next-word precition task (Reddit).

