# OpenReview forum: "FedSynth: Gradient Compression via Synthetic Data in Federated Learning"
_NeurIPS.cc/2022/Workshop/Federated_Learning — FL-NeurIPS 2022 Poster_

### Official Review · Reviewer_dsfz · 2022-10-15

In this paper, the authors propose FedSynth to reduce communication costs in federated learning (FL). In particular, instead of communicating the model updates from clients to the server, in FedSynth, the clients generate a synthetic dataset such that when the synthetic dataset Is used for local training, it performs well on the true local training data. The clients, then, send their synthetic dataset to the server. The authors state that this synthetic dataset is small and requires a lower cost than communicating the model updates (i.e., gradients). Empirically, they show that FedSynth outperforms Random Masking in some datasets under some of the compression rates.

- In terms of the technical contributions, I don't see a significant difference from the prior work [1], which the authors cite.

- The only noteworthy difference from [1] seems to be the additional experimental results on MNIST, FedMNIST, and Reddit datasets, comparing FedSynth against Random Masking. The authors never explain or give any reference to the only baseline method Random Masking. While I have a guess as to what Random Masking might be doing, the paper needs a detailed description of Random Masking and the implementation details. For instance, it is not clear how the compression rate for Random Masking is calculated.

- Looking at the experimental results in Table 1, we can't say that FedSynth is outperforming the only baseline Random Masking consistently. In many configurations, Random Masing has higher accuracy. Having lower performance occasionally is, of course, okay when the comparisons against recent and relevant baselines are extensive. However, this is not true for this paper. There is only one baseline, for which no description is provided. So, it's hard to say anything about the empirical performance of FedSynth compared to the existing methods.

- Also, I don't understand why the authors compare only against Random Masking. Among the sparsification methods, Top-k [2], rTop-k [3], or SuRP [4]; and among the quantization methods, QSGD [5] or DRIVE [6] are very relevant baselines. I believe, it is hard to justify how much communication gain FedSynth provides without additional baselines, given that we also don't know any details about the present baseline Random Masking. I think including some of these baselines (or other recent baselines) is necessary for a more justified empirical evaluation. The paper also misses most of these works in the related work section.

- Lastly, the writing needs some improvement. There were broken sentences such as "Although similar to our approach..." sentence in line 55, typos such as "to the them" in line 65, and missing references and explanations.

[1] Goetz, Jack, and Ambuj Tewari. "Federated learning via synthetic data." arXiv preprint arXiv:2008.04489 (2020).

[2] Lin, Yujun, et al. "Deep gradient compression: Reducing the communication bandwidth for distributed training." arXiv preprint arXiv:1712.01887 (2017).

[3] Barnes, Leighton Pate, et al. "rTop-k: A statistical estimation approach to distributed SGD." IEEE Journal on Selected Areas in Information Theory 1.3 (2020): 897-907.

[4] Isik, Berivan, Tsachy Weissman, and Albert No. "An information-theoretic justification for model pruning." International Conference on Artificial Intelligence and Statistics. PMLR, 2022.

[5] Alistarh, Dan, et al. "QSGD: Communication-efficient SGD via gradient quantization and encoding." Advances in neural information processing systems 30 (2017).

[6] Vargaftik, Shay, et al. "DRIVE: one-bit distributed mean estimation." Advances in Neural Information Processing Systems 34 (2021): 362-377.

---

### Official Review · Reviewer_w1ws · 2022-10-17
**Paper with a novel idea**

This paper presents fedsynth, a new gradient compression approach.

In this approach, instead of transmitting the gradients or compressed gradients,  fedsynth transmits modified synthetic data for synchronization.

The idea is novel. It suggests a new way of federated optimization.

Questions:

(1) How do we initialize the synthetic data?

(2) Is there any theoretical guarantees for recovery if we simplify the models to linear?

(3) How does fedsynth compare with the topk gradient sparsification or low-rank approximation?

---

### Official Review · Reviewer_fPN2 · 2022-10-18
**A good paper that could use some stronger empirical baselines**

I will preface this with saying that I enjoyed this paper a lot, and that I think this should be accepted. The only reason I want to qualify this review in this specific way is so that any weaknesses of the paper are not interpreted as ambiguity in my rating, but as constructive feedback for any future revisions to this work.

At its core, this paper introduces a simple idea (essentially, learn local datasets instead of performing local training) that raises a host of interesting questions and potential follow-up ideas. Moreover, the motivation is relatively clear; Synthetic datasets have the potential to be orders of magnitude smaller than models, and so clients should learn datasets such that optimizing the model at the learned dataset produces an updated model that is good on the client's sensitive data (quite a mouthful). I enjoy that this marries a kind of re-examination of FL itself with the ability of modern auto-differentiation systems to easily change exactly what is being learned.

If anything, I think the authors could discuss further implications of federated distillation. For example, can such techniques reduce the computation of clients? I would assume that local compute is roughly the same (since model forward passes still need to be computed) but perhaps the backward pass savings are substantial (since we are only updating the data). I would also note that the objective in (3) and (4) are intimately related to meta-learning methods (eg. MAML), something that could be explored as well.

My feedback to the authors would be that the empirical baselines in the paper are relatively weak. The models trained are relatively small, small enough to warrant the question of whether someone would need to do dataset distillation for such models. Similarly, the FEMNIST benchmarks seem somewhat low. While the client split is different than that of the FEMNIST in [Reddi et al., 2021] (for example), an accuracy of < 70% is not particularly high. As such, I would recommend that the authors use slightly larger and more interesting models, in order to get 1) slightly more robust and accurate baselines and 2) so that the models in question are large enough that dataset distillation may be preferable. Similarly, the compression baseline of random masking is relatively weak. Generally, methods like QSGD [Alistarh et al.] are much stronger, while still being relatively easy to implement.

### Minor comments/questions

First, how is the differentiation actually being done for the Reddit next-word prediction task? Generally stacked LSTM models will use discrete tokenization, in which case continuous differentiation cannot be applied directly. Discussing in greater detail some of these implementation details would be highly beneficial.

Second, why does the server reconstruct every client model individually (Algorithm 1, Line 12) instead of amalgamating the datasets and simply learning over the entire dataset? Intuitively, the latter allows for random shuffling across datasets, which effectively resolves issues of heterogeneity which would still be present when learning a bunch of local models and averaging them (Line 13).

---

### Decision · Program_Chairs · 2022-10-20

Accept (Poster)